# Involvement of *CYP51A* and *CYP51B* in Growth, Reproduction, Pathogenicity, and Sensitivity to Fungicides in *Colletotrichum siamense*

**Shuodan Hu** [1,†]**, Jianyan Wu** [1,†]**, Xiaoqi Yang** [1]**, Wenfei Xiao** [2]**, Hong Yu** [2] **and Chuanqing Zhang** [1,*]

1    College of Advanced Agricultural Sciences, Zhejiang A&F University, Lin'an, Hangzhou 311300, China
2    Research Institute for the Agriculture Science of Hangzhou, Hangzhou 310013, China
*    Correspondence: cqzhang@zafu.edu.cn
†    These authors contributed equally to this work.

**Abstract:** Strawberry crown rot is a serious fungal disease that poses a great threat to strawberry production in the growth cycle. The dominant pathogens of strawberry crown rot pathogens were different in different periods. The main pathogen of strawberry crown rot at the seedling stage is unclear. In this study, 74 *Colletotrichum* spp. were isolated from 100 strawberry plants at the seedling stage. Based on the morphological observations and phylogenetic analysis of multiple genes (*ACT*, *CAL*, *CHS*, *GAPDH*, and ITS), all 74 tested isolates were identified as *C. gloeosporioides* species complex, including 69 isolates of *C. siamense* and 5 isolates of *C. fructicola*. *Colletotrichum siamense* is the main pathogen of strawberry crown rot at the seedling stage in Zhejiang, China. The sterol demethylation inhibitors (DMIs) were used to control strawberry crown rot, and their target was the *CYP51* gene. The role of the homologous *CYP51* gene in growth, reproduction, pathogenicity, and sensitivity to DMI fungicides in *C. siamense* has not been determined. Our study found that the pathogenicity of *CsCYP51A* deletion mutants to strawberry leaves and stems was weakened. The hyphae growth rate of *CsCYP51B* deletion mutants was significantly slower than that of the wild type, but the sporulation and appressorium production rates increased. *CsCYP51B* deletion mutants had significantly increased pathogenicity to the stem. Deletion of *CsCYP51A* led to increased sensitivity to prothioconazole, ipconazole, hexaconazole, triadimefon, prochloraz, tebuconazole, metconazole, propiconazole, and difenoconazole. *CsCYP51B* deletion mutants were more insensitive. Our results indicate that the effect of the homologous *CsCYP51* gene on hyphae growth, pathogenicity, and sensitivity to DMI fungicides differs.

**Keywords:** *C. siamense*; *CYP51A*; *CYP51B*; DMI fungicides; mutants

## 1. Introduction

Strawberry (*Fragaria* × *ananassa* Duch.) is a small berry and has the largest planted and cultivated area in the world [1]. In recent years, strawberry crown rot has become a destructive disease in strawberry production. When the vascular bundle is infected, it becomes brown; strawberries eventually wilt due to the inability to absorb water and nutrients, causing considerable economic damage [2]. The main pathogens of strawberry crown rot reported worldwide are the *Colletotrichum gloeosporioides* complex species and *C. acutatum* complex species [3–5]. In China, *C. gloeosporioides* complex species is the main pathogens of strawberry crown rot [6]. However, the dominant pathogen of strawberry crown rot pathogens were different in different periods [7]. *C. siamense* is the main pathogen of strawberry crown rot in transplanting period. *C. siamense* belong to *C. gloeosporioides* complex species and is important hemibiotrophic fungus pathogen that affects a wide range of crops, including fruit and vegetables, and can infect the leaves, stems, fruit, and other parts of host plants [8]. The main pathogen of strawberry crown rot in the Zhejiang Province at the seedling stage is unclear.

The sterol demethylation inhibitors (DMIs) are widely used to control strawberry crown rot in production. DMI fungicides disturb fungal growth by inhibiting cytochrome 14-alpha-demethylases (*CYP51*) [9]. At present, the mechanism of resistance of pathogenic fungi to DMI fungicides has been reported mainly in three ways: (i) *CYP51* gene point mutation reduces the affinity between the fungicide and the target protein [10,11], (ii) overexpression of the *CYP51* gene [12,13], and (iii) increased efflux pump activity [14]. With the extensive use of DMIs, *Colletotrichum* spp. have become less sensitive to DMI fungicides and have developed resistance over the past few decades [15,16]. However, the mechanism by which *Colletotrichum* spp. are resistant to DMI fungicides is unclear.

Most ascomycetes only carry two copies of *CYP51* gene: *CYP51A* and *CYP51B* [17]. Thus far, *CYP51C* has only been reported in *Fusarium* spp. [18]. The effects of the homologous *CYP51* gene on hyphae growth, reproduction, pathogenicity, and sensitivity to DMI fungicides have been previously reported. The *CYP51* deletion mutants of *Fusarium* spp. had no effect on hyphae growth and sensitivity to DMI fungicides [19,20]. Similarly, the role of the homologous *CYP51* gene may differ among *Colletotrichum* spp. [21]. In *C.gloeosporioides*, *CYP51A* deletion mutants led to attenuated growth on PDA medium, whereas *CYP51B* deletion caused enhanced growth. The deletion mutants of *CYP51A* and *CYP51B* failed to produce spores and showed decreased virulence. *CYP51A* deletion mutants became more sensitive to DMI fungicides, but not *CYP51B* deletion mutants [22]. *CYP51A* deletion mutants of *C. nymphaeae* have increased sensitivity to propiconazole, diniconazole, prothioconazole, cyproconazole, epoxiconazole, flutriafol, and prochloraz. However, *CYP51A* deletion mutants of *C. fioriniae* did not change their sensitivity to prochloraz. Hyphae growth and virulence of *CYP51* deletion mutants were not changed in *C. nymphaeae* or *C. fioriniae* [23].

The function of two homologous *CYP51* genes in *C. siamense* remains unclear. The objective of this study was to investigate the effects of *CYP51A* and *CYP51B* genes on hyphae growth, reproduction, pathogenicity, and sensitivity to DMI fungicides in *C. siamense*. The findings of this study will improve our understanding of the biology of *CYP51* genes in fungi.

## 2. Materials and Methods

### 2.1. Isolation of Colletotrichum spp.

Crown rot samples were collected from plants with a brown vascular bundle and wilting from 10 greenhouses in Jiande, Zhejiang Province (27°02′ to 31°11′ N, 118°01′ to 119°28′ E) from July to September 2021. The diseased tissues from the infected plants were washed with tap water, cut into 5 × 5 mm pieces using sterilized scissors, soaked in 75% alcohol for 30 s, soaked in 3% sodium hypochlorite solution for 1 min, washed with sterile distilled water three times, and dried on sterile filter paper. Each tissue piece was placed on a plate containing potato dextrose agar (PDA) medium supplemented with kanamycin sulfate and streptomycin sulfate (100 mg/L) and incubated at 28 °C. After 3–5 days of incubation, the mycelia were transferred to a new PDA plate [24,25]. The single-conidium isolates were stored on PDA slants at 4 °C.

### 2.2. Morphological Characterization

Morphological and cultural characterizations were performed according to previously described methods [26]. Mycelia plugs (5 mm in diameter) were taken from the edges of 5-day-old colonies of tested isolates and transferred to new PDA plates. The mycelium diameter was measured and the growth rate was calculated after incubation at 28 °C for 7 days. The colony morphology and color of aerial mycelia were recorded [26,27]. Each isolate was repeated 3 times. Moreover, the shape and color of the conidia and appressoria were observed using a light microscope, and 50 conidia and appressoria were randomly selected and measured to determine their size by ZEN V.6.0 software (Zeiss, Jena, Germany).

### 2.3. Molecular Identification and Phylogenetic Analysis

The total DNA of each tested isolate was extracted using a fungi genomic DNA rapid extraction kit (B518229-0100; Sangon Biotech, Shanghai, China). The actin (*ACT*), calmodulin (*CAL*), chitin synthase (*CHS*), glyceraldehyde 3-phosphate dehydrogenase (*GAPDH*) and internal transcribed spacer (*ITS*) genes were amplified using the primer pairs ACT-512F/ACT-783R [28], CL1C/CL2C [29], CHS-79F/CHS-354R [30], GDF/GDR [31], and ITS-1F/ITS4 [32]. Molecular identification was performed according to the method described previously [33]. The referenced standard isolates used in this study are listed in Supplementary Table S1. *Colletotrichum boninense* (CBS: 123755) was used as the out-group [33]. All sequences were compared and corrected in MEGA 5.0 [34]. The modified sequences were concatenated in Sequence Martix 1.8. Modeltest3.7.win, Win paup4b10-console, and Mrmodeltest2, as implemented in MrMTgui, were used to estimate the best model of nucleotide substitution [35]. Bayesian inference (BI) phylogenies were constructed using Mr. Bayes v. 3.1.2 [36]. Six simultaneous Markov chains were run for 300,000 generations each. Phylogenetic trees were drawn using treeView [37]. The alignments and trees were deposited in treeBase.

### 2.4. Construction of CsCYP51A and CsCYP51B Deletion Mutants

The double-joint (DJ) PCR approach [38] was used to generate the gene replacement construct for each target gene. Briefly, two primer pairs (CYP51a-UP-F/CYP51a-UP-R and CYP51a-DOWN-F/CYP51a-DOWN-R) (Table 1) were used to amplify the upstream and downstream sequences of *CsCYP51A* from the genome of the wild-type JD-A-12 strain. The primers HPH-F/HPH-R were used to amplify a 1349-bp fragment encoding the HPH cassette containing the hygromycin-resistant gene and the trpC promoter. The three amplicons (upstream, HPH cassette, and downstream) were fused by a second round of DJ PCR. Based on the fused fragment, the final PCR products with an overlapping part of 3188 and 3090 bp were amplified by the nested primer pair CYP51a-Nest-F/CYP51a-Nest-R (Table 1). The *CsCYP51B* deletion mutants were constructed using the same protocols.

**Table 1.** Primers used in this study.

| Primers | Direction | Length (bp) | Sequence (5′→3′) |
|---|---|---|---|
| CYP51a-UP-F | Forward | 987 | GGAGTCCTCGAATCTGAGTTC |
| CYP51a-UP-R | Reverse | | AAAATAGGCATTGATGTGTTGACTCCCTCGG AAGTTCTATGCCTTC |
| CYP51a-DOWN-F | Forward | 1017 | CTCGTCCGAGGGCAAAGGAATAGAGTAGCTGATGGCGACATGAACCGTG |
| CYP51a-DOWN-R | Reverse | | CATGCTGGCAACGGAAGTG |
| CYP51a-ID-F | Forward | 1649 | GGAAGCCATTATATGAGAAG |
| CYP51a-ID-R | Reverse | | CATGCTGGCAACGGAAGTG |
| CYP51a-Nest-F | Forward | 3188 | GGTGTCCATCTAAGGAATTGG |
| CYP51a-Nest-R | Reverse | | CATGCTGGCAACGGAAGTG |
| CYP51b-UP-F | Forward | 949 | GCAATTGCGAGCATGTGAGTG |
| CYP51b-UP-R | Reverse | | AAAATAGGCATTGATGTGTTGACCTCCGCTGGTAGTGTGAAGGGAAG |
| CYP51b-DOWN-F | Forward | 1016 | CTCGTCCGAGGGCAAAGGAATAGAGTAGGGGAATGTATATTGTAAGCC |
| CYP51b-DOWN-R | Reverse | | CTTCTGCATCATGAGCTGGAC |
| CYP51b-ID-F | Forward | 1541 | CTCTCTCGCGCCACTGCTG |
| CYP51b-ID-R | Reverse | | GTGATGTCATAACGTCTTTTG |
| CYP51b-Nest-F | Forward | 3096 | CTAGCGAATCGAAGACGGAG |
| CYP51b-Nest-R | Reverse | | GCGCCGTCGACTCAGGGTAGG |
| HPH-F | Forward | 1349 | GGAGGTCAACACATCAATGCCTATT |
| HPH-R | Reverse | | CTACTCTATTCCTTTGCCCT |

### 2.5. Transformation of C. siamense

Protoplast transformation of *C. siamense* was carried out using a previously described protocol [38,39]. The mycelia on the edge of the colony was crushed and placed in 30 mL YEPD medium and shaken at 180 rpm for 36 h at 28 °C. To prepare protoplasts, mycelia were collected by filtration and washed with 0.7 M NaCl buffer. The cell wall lysate contained 0.3 g cellulase, 0.3 g lysozyme, and 0.1 g collapse enzyme in 10 mL of 0.7 M NaCl buffer as the enzyme mixture, which was filtered through a 0.22-μm MILLEX GP sterile filter membrane.

A flask with 10 mL of filter-sterilized enzyme mixture and 1 g of mycelia was incubated at 100 rpm for 3 h at 28 °C. The resulting protoplasts were filtered through three layers of lens cleaning tissue and collected by centrifugation at 5000 rpm for 5 min. They were then washed twice in 1 mL of sorbitol-Tris-calcium (STC) buffer. The protoplasts were precipitated with 750 μL STC solution, and the final PCR product, heparin sodium and SPTC were added in turn, mixed and left on ice for 30 min. SPTC was added, left to stand for 20 min and transferred to 20 mL RM medium at 28 °C and 100 rpm for 12 h. RM medium was added to 200 μg/mL hygromycin modified with PDA overnight. After 2–4 days of incubation, hygromycin-resistant colonies (transformants) were transferred onto fresh PDA medium amended with 200 μg/mL hygromycin for two generations to confirm resistance. The total DNA of the transformants was extracted using a fungi genomic DNA rapid extraction kit (B518229-0100; Sangon Biotech, Shanghai, China). Putative gene deletion mutants were identified using PCR assays with relevant primers (CYP51a-ID-F/CYP51a-ID-R, CYP51b-ID-F/CYP51b-ID-R) (Table 1).

### 2.6. Phenotype Analysis

The mycelia plugs (5 mm in diameter) were transferred to the new PDA plates and incubated at 28 °C. After 7 days, the hyphae growth rate, sporulation and appressorium production rate were calculated, and colony morphology was described [26,27]. For each deletion genotype, three mutants were measured. For appressorial formation assays, the conidial suspension was adjusted to a concentration of $1 \times 10^6$ conidia/mL, and 10 μL was placed onto a plastic cover slip (Thermo Fisher Scientific, Waltham, MA, USA) and incubated at 28 °C for 16 h. Conidial germination and appressorial formation rates were quantified microscopically.

### 2.7. Pathogenicity Assays

We tested the pathogenicity of each deletion genotype to the leaves and stems of 8-week-old strawberry. Before inoculation, the leaves and stems were surface sterilized by dipping in 1% sodium hypochlorite for 30 s and then in 70% ethanol. They were then washed three times with distilled water and left to dry on sterile paper. The wounded condition was determined using the pin-pricking method [40]. Depending on the size of the leaves, a sterile sharp needle was used to prick 4–8 wounds on the adaxial surface of each leaf. For the stem, 5 wounds were pricked in the middle of each 8-cm stem. Spores were collected and adjusted to $1.0 \times 10^6$ conidia/mL using a hemocytometer. Three mutants were inoculated for each deletion genotype, and three leaves and stems of each mutant were inoculated. Sterile water was used for inoculation as a blank control. The diameter of the lesions on the leaves and stems was measured 3 days after inoculation. One-way ANOVA was performed to analyze the significance difference using IBM SPSS Statistics V22 software, and the least significant difference (LSD) test was applied to separated mean values for different species in the pathogenicity test at $p = 0.05$ level.

### 2.8. Determination of Sensitivity to DMI Fungicides

To evaluate the sensitivity of *CsCYP51A* and *CsCYP51B* mutants to nine DMI fungicides, namely prothioconazole, ipconazole, hexaconazole, triadimefon, prochloraz, tebuconazole, metconazole, propiconazole, and difenoconazole, we first measured the $EC_{50}$ of wild-type JD-A-12 to nine DMI fungicides according to the mycelial growth rate method [41]. Each active component was added at concentrations of 0, 0.3125, 0.625, 1.25, and 2.5 μg/mL in the microbicide-modified PDA plates. The $EC_{50}$ value was used as the concentration to determine the sensitivity of each deletion genotype to nine DMI fungicides. The mycelia plugs (5 mm in diameter) were transferred to the PDA medium treated with different concentrations of fungicides, and the wild type was used as the control. Three mutants were tested for each deletion genotype, and each mutant was repeated on three plates. After being cultured at 28 °C for 7 days, the colony diameter was determined, and the inhibition rate was calculated.

## 3. Results

### 3.1. Isolation and Identification of Colletotrichum spp.

A total of 74 Colletotrichum spp. were isolated from 100 strawberry plants. Pathogenicity was determined according to Koch's postulate, and all isolates were pathogenic. According to morphological and cultural characterization, all 74 isolates were grouped into two Colletotrichum species: *C. siamense* and *C. fructicola*. The proportion of *C. siamense* was 93.24%.

Colletotrichum siamense colonies were medium gray, and the back surface was gray-green in the middle and white at the edges. Colletotrichum fructicola colonies were gray-green in the middle and white at the edges (Figure 1). The two Colletotrichum species had similar conidia and appressorium morphology. The spores and appressoria of *C. fructicola* were larger than those of *C. siamense* (Table 2). There were no significant differences in mycelial growth rates. Phylogenetic trees were constructed using combined ACT, GADPH, CHS, CAL and ITS datasets consisting of 74 Colletotrichum isolates with *C. boninense* (CBC 123755) as the outgroup taxa. The concatenated alignment included 1356 characters. The boundaries of the loci used in the alignment were as follows: ACT: 1–152; CAL: 153–549; CHS: 550–761; GADPH: 762–965; and ITS: 966–1356. The GenBank no. of tested isolates are listed in Supplementary Table S1. All isolates were identified as *C. gloeosporioides* species complex and fell into two clades, with 69 isolates clustered in *C. siamense* and 5 isolates clustered in *C. fructicola* (Figure 2).

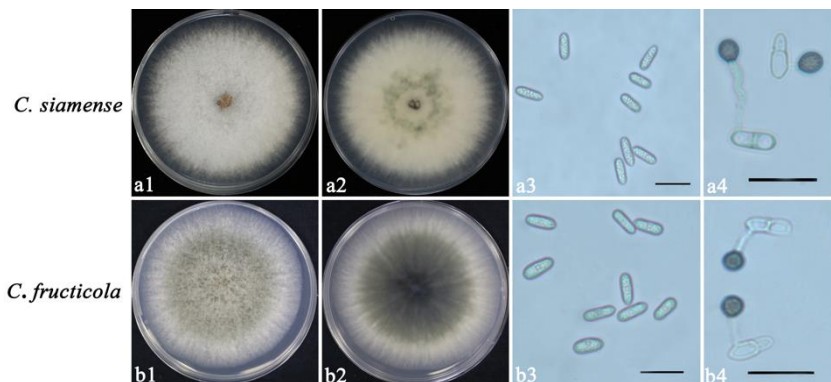

**Figure 1.** Colony morphology of Colletotrichum siamense and *C. fructicola* from the top of the PDA plate (**a1–b1**), from the underside of the PDA plate (**a2–b2**), conidia (**a3–b3**), and appressoria (**a4–b4**), scale bar: (**a3–b3, a4–b4**) = 20 μm.

**Table 2.** Size of spores and appressoria, growth rates, and sporulation of Colletotrichum siamense and *C. fructicola*.

| Species | Strain Number | Conidia [y] | | Appresoria [y] | | Growth Rate (mm/Day) [y] | Sporulation ($\times 10^6$) [y] |
| | | Length (μm) | Width (μm) | Length (μm) | Width (μm) | | |
|---|---|---|---|---|---|---|---|
| *C. siamense* | JD-A-12 | 14.17 ± 0.63 [b] | 5.64 ± 0.46 [b] | 6.97 ± 0.45 [b] | 6.36 ± 0.70 [ab] | 13.54 ± 0.37 [a] | 21.06 ± 5.71 [a] |
| | JD-A-26 | 15.57 ± 0.21 [b] | 6.87 ± 0.19 [a] | 7.10 ± 0.25 [b] | 5.69 ± 0.66 [ab] | 13.42 ± 0.08 [a] | 20.20 ± 2.44 [ab] |
| *C. fructicola* | JD-A-14 | 17.14 ± 0.59 [a] | 6.21 ± 0.19 [a] | 8.19 ± 0.15 [a] | 6.83 ± 0.43 [b] | 13.48 ± 0.04 [a] | 16.36 ± 3.72 [b] |
| | JD-A-22 | 16.68 ± 0.30 [a] | 6.63 ± 0.13 [a] | 8.90 ± 0.72 [a] | 7.03 ± 0.19 [a] | 13.51 ± 0.07 [a] | 14.53 ± 3.15 [b] |

[y] Data are the mean ± standard error. Mean values with the same letters were not statistically different ($p > 0.05$) according to the least significant difference (LSD) test.

### 3.2. Deletion of CsCYP51 in C. siamense

Three CsCYP51A gene deletion mutants were obtained and identified using PCR analysis with the primer pair CYP51a-ID-F/CYP51a-ID-R. This primer pair amplified 1649-bp fragments from the CsCYP51A mutants (Figure 3B). Three CsCYP51B gene deletion mutants were obtained. The primer pair CYP51b-ID-F/ CYP51b-ID-R amplified 1541 bp from CsCYP51B (Figure 3D).

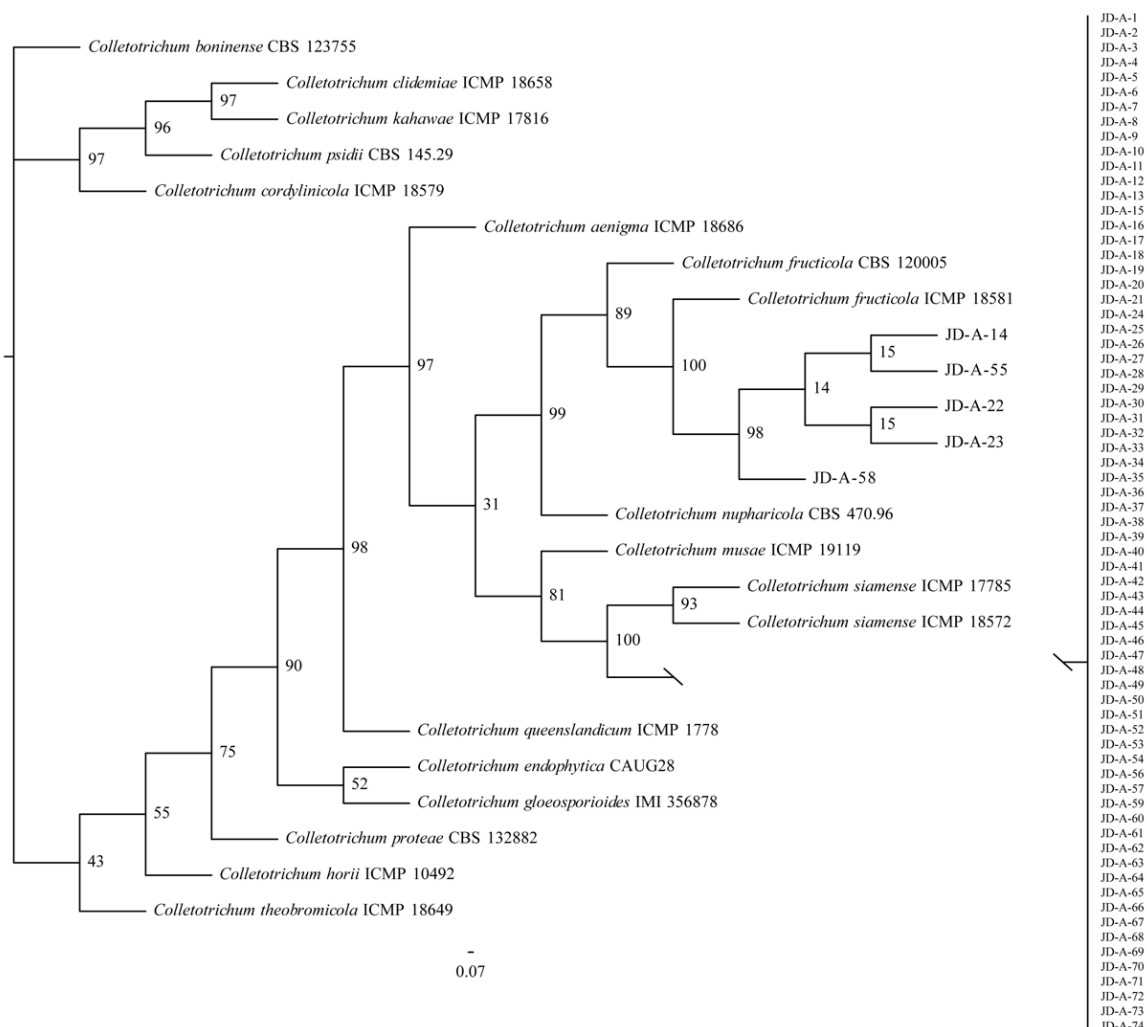

**Figure 2.** Bayesian inference phylogenetic tree of Colletotrichum gloeosporioides species complex isolated from strawberry. The tree was constructed based on ACT, CAL, CHS, GAPDH and ITS genes. Colletotrichum boninense was used as an outgroup. The scale bar shows 0.07 expected changes per site.

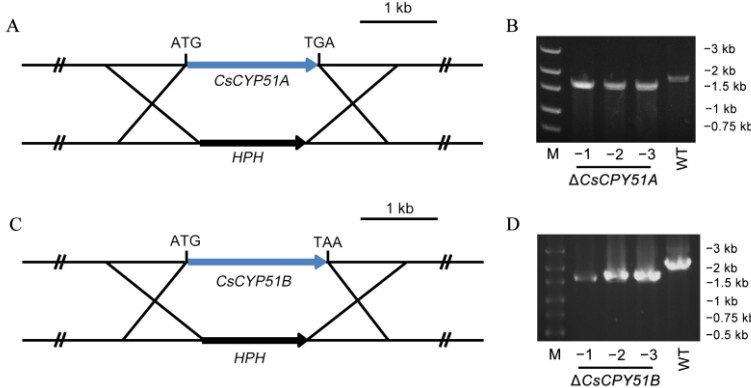

**Figure 3.** Generation and identification of CsCYP51A and CsCYP51B deletion mutants by gene replacement. (**A**) Schematic representation of the CsCYP51A replacement strategy. (**B**) PCR verification of the CsCYP51A deletion mutation. (**C**) Schematic representation of the CsCYP51B replacement strategy. (**D**) PCR verification of the CsCYP51B deletion mutation.

### 3.3. Biological Characteristics of CsCYP51 Mutants

The colony morphology of the CsCYP51A deletion mutants was not significantly different from that of wild-type JD-A-12 on PDA plates. The mycelium was gray-white, and the back surface was gray-green in the middle and gray-white at the margin (Figure 4). The hyphae of the CsCYP51B deletion mutant were denser and produced orange-red precipitates on the PDA plate (Figure 4). The growth rates of CsCYP51A and CsCYP51B deletion mutants were significantly lower than that of wild-type JD-A-12, and the growth rates of the two mutants were $10.39 \pm 0.10$ mm/d and $7.28 \pm 0.09$ mm/d, respectively. Sporulation and the appressorium production rate of CsCYP51B deletion mutants were significantly higher than those of the wild-type JD-A-12 (Table 3).

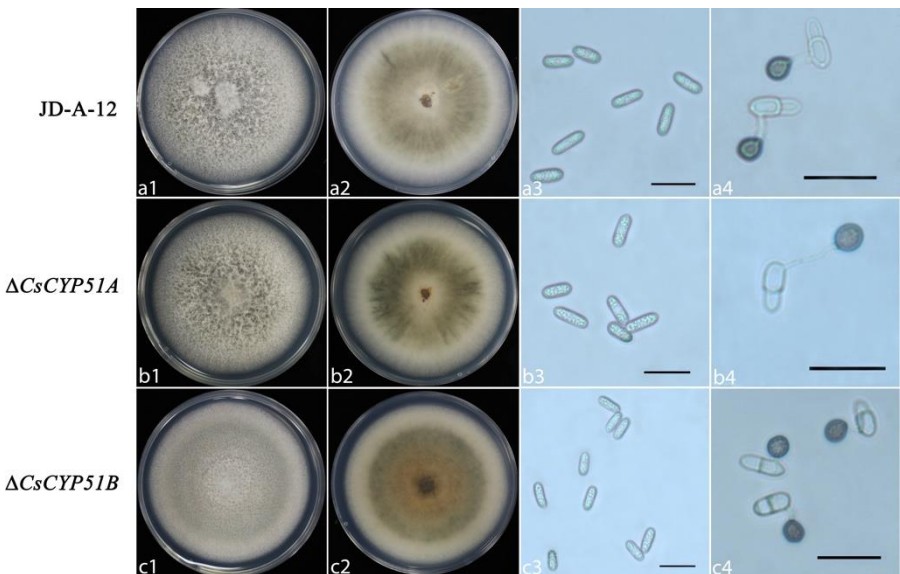

**Figure 4.** Colony morphology of wild-type JD-A-12 and CsCYP51A and CsCYP51B deletion mutants from the top of the PDA plate (**a1–c1**, from the underside of the PDA plate (**a2–c2**), conidia (**a3–c3**) and appressoria (**a4–c4**), scale bar: (**a3–c3**, **a4–b4**) = 20 μm.

**Table 3.** The growth rate, sporulation, and appressorium production rate of wild-type JD-A-12 and CsCYP51A and CsCYP51B deletion mutants.

| Species | Growth Rate (mm/Day) [y] | Sporulation ($\times 10^6$) [y] | Appressorium Production Rate (%) [y] |
|---|---|---|---|
| JD-A-12 | $11.56 \pm 0.07$ [a] | $1.3 \pm 0.09$ [b] | $13.82 \pm 1.74$ [b] |
| ΔCsCYP51A | $10.39 \pm 0.10$ [b] | $1.7 \pm 0.13$ [b] | $14.47 \pm 1.22$ [b] |
| ΔCsCYP51B | $7.28 \pm 0.09$ [c] | $4.33 \pm 0.30$ [a] | $17.80 \pm 1.25$ [a] |

[y] Data are the mean $\pm$ standard error. Mean values with the same letters were not statistically different ($p > 0.05$) according to the least significant difference (LSD) test.

### 3.4. Effect of CYP51 on Pathogenicity

The disease spot diameter was measured 3 days after inoculation. CsCYP51A deletion mutants significantly reduced the pathogenicity to the stem and leaves (Figure 5), and the lesion diameters were $3.4 \pm 0.4$ mm and $1.9 \pm 0.2$ mm, respectively. The CsCYP51B deletion mutants significantly increased pathogenicity to the stem, with a diameter of $10.8 \pm 0.6$ mm. However, there was no significant difference in leaf pathogenicity between the wild type and CsCYP51B deletion mutants (Figure 5).

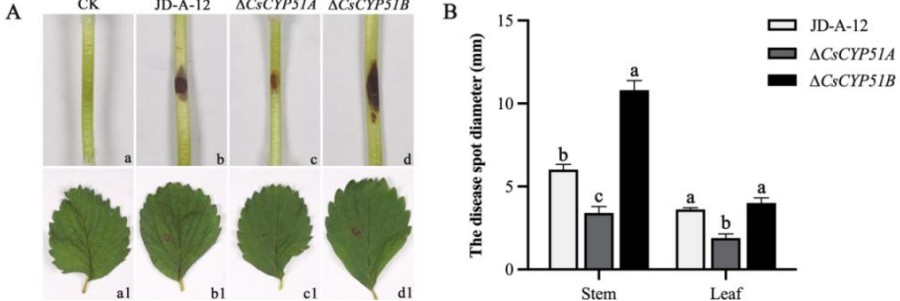

**Figure 5.** Pathogenicity of CsCYP51A and CsCYP51B mutants to strawberry leaves and stems. (**A**) JD-A-12, CsCYP51A, and CsCYP51B caused disease spots on the stem and leaf. (**B**) Pathogenicity difference analysis of JD-A-12, CsCYP51A, and CsCYP51B. Values with the same letters were not statistically different ($p > 0.05$) according to the least significant difference (LSD) Test.

### 3.5. Effects of CYP51 Gene Deletion on the Sensitivity of C. siamense to DMIs

The $EC_{50}$ values of wild-type JD-A-12 for prothioconazole, ipconazole, hexaconazole, triadimefon, prochloraz, tebuconazole, metconazole, propiconazole, and difenoconazole were 2.27, 0.1, 1.07, 19.26, 0.2, 0.85, 0.36, 0.36, and 0.47 µg/mL, respectively. According to the $EC_{50}$ values, we determined the sensitivity of the CsCYP51A and CsCYP51B deletion mutants to nine DMI fungicides. The mycelium growth on the PDA plate was shown in Figure 6A. CsCYP51A deletion mutants were more sensitive to all tested DMI fungicides. Compared with the wild type JD-A-12, the inhibition rate of nine DMI fungicides to *CsCYP51A* deletion mutants was significantly increased (Figure 6B), especially for hexaconazole and triadimefon, whose inhibition rates reached 86.60% and 91.06%. The growth rates of CsCYP51B deletion mutants were lower than that of wild type JD-A-12. According to the results of statistical analysis, the inhibition rate of nine DMI fungicides to CsCYP51A deletion mutants was significantly reduced compared with the wild type (Figure 6B). This means that CsCYP51B deletion mutants were less sensitive to the nine DMI fungicides than wild-type JD-A-12.

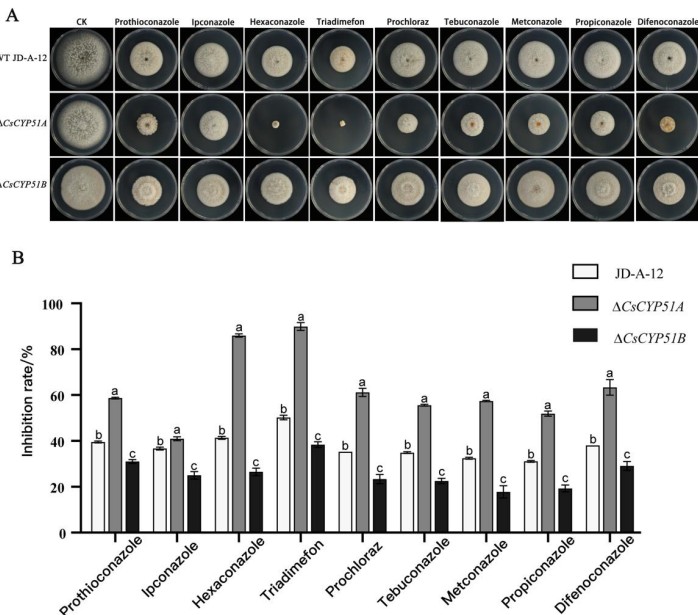

**Figure 6.** Using the $EC_{50}$ value as concentration to determine sensitivities of wide type JD-A-12, CsCYP51A and CsCYP51B deletion mutants to nine DMI fungicides. (**A**) The mycelium growth on a PDA plate. (**B**) Inhibition rate of nine DMI fungicides to CsCYP51A and CsCYP51B deletion mutants. Values with the same letters were not statistically different ($p > 0.05$) according to the least significant difference (LSD) Test.

## 4. Discussion

Strawberry crown rot caused by *Colletotrichum* spp. causes serious economic losses [6]. Based on morphological and phylogenetic analyses, 74 isolates of *Colletotrichum* spp. were isolated from 100 infected strawberry plants at the seedling stage in Jiande, Zhejiang Province. All *Colletotrichum* spp., namely *C. siamense* and *C. fructicola* belonged to the *C. gloeosporioides* species complex, in which *C. siamense* was the main species, accounting for 93.24%. It is reported that *C. siamense* and *C. fructicola* were the most prevailing pathogens causing strawberry crown rot in the Asia–Pacific region [42]. Our research also confirms this. Of course, other *Colletotrichum* species also can cause strawberry crown rot [43,44].Our study indicates that *C. siamense* was the dominant species of strawberry crown rot at different periods in Zhejiang province [7]. There was no significant difference in spore morphology and mycelial growth rate between *C. siamense* and *C. fructicola;* the spores and appressoria of *C. fructicola* were larger than those of *C. siamense*. This is also consistent with previous research [45].

*C. siamense* is the main pathogen of strawberry crown rot at the seedling stage in Zhejiang Province. This disease is mainly controlled by fungicides, such as DMI, whose target gene is *CYP51*. With the extensive use of DMI fungicides, the sensitivity of *C. siamense* decreased and resistance was developed [19,46]. Therefore, we explored the involvement of *CYP51* with growth, reproduction, pathogenicity, and sensitivity to fungicides in *C. siamense*. In this study, *CsCYP51A* deletion mutants had no significant effect on hyphae growth and sporulation. The *CsCYP51B* deletion mutants significantly slowed the mycelial growth rate but did not affect the production of spores and appressoria. Sporulation and appressorium production rates were significantly higher than those of wild-type JD-A-12. This suggests that neither *CYP51A* nor *CYP51B* is necessary for fungal growth [47]. *CYP51B* deletion in *C. nymphaeae* and *C. fioriniae* had no effect on hyphae growth, which further indicated that CYP51B was not necessary for the growth of *Colletotrichum* spp. [48]. The *CsCYP51A* deletion mutant had reduced pathogenicity to stems and leaves, while the *CsCYP51B* deletion mutant had increased pathogenicity to stems. In *C. gloeosporioides*, *CYP51A* deletion mutant also reduced pathogenicity [22]. *CYP51A* may play an important role in the infection process [26]. The differential pathogenicity of the *CsCYP51B* deletion mutant on stems and leaves may be due to host site specialization in strawberry plant infection [42]. The pathogenicity of *Colletotrichum* spp. is closely related to appressorium production, which may also be associated with an increased rate of appressorium production in *CsCYP51B* deletion mutants [49]. The *CYP51B* gene may negatively regulate pathogenicity in *C. siamense*.

Different DMI fungicides have different binding affinities with the *CYP51* protein due to their different chemical structures [50]. In this study, we determined the sensitivity of *CsCYP51A* and *CsCYP51B* deletion mutants to nine DMI fungicides. *CsCYP51A* deletion mutants were more sensitive to the nine DMI fungicides and *CsCYP51B* deletion mutants was insensitive. In other words, *CYP51A* in *C. siamense* may be the main target for these nine DMI fungicides. This phenomenon has also been observed in *C. gloeosporioides* [22]. *CYP51B* deletion mutants showed similar sensitivity to nine DMI fungicides. The *CYP51A* gene was found to be more variable than *CYP51B* [51]. The conservation of *CYP51B* may explain why *CYP51B* deletion mutants are mostly similarly sensitive to all tested fungicides. When two fungicides with different primary targets are used in combination, a synergistic effect can be achieved in disease control [48]. We can also continue to screen DMI fungicides targeting *CYP51B*. In our study, the expression and function of CYP51 gene in the mycelium growth, sporulation, and pathogenicity of *C. siamense* need to be further verified. In general, the *CYP51* gene in *C. siamense* differentially affected growth, reproduction, pathogenicity, and sensitivity to DMI fungicides. Our findings provide novel insights into understanding the resistance mechanism to DMIs.

**Supplementary Materials:** The following supporting information can be downloaded at: https://www.mdpi.com/article/10.3390/agronomy13010239/s1, Table S1: *Colletotrichum* spp. used in multi-gene analysis in this study.

**Author Contributions:** Conceptualization, J.W. and C.Z.; methodology, J.W., H.Y., and C.Z.; software, S.H. and X.Y; validation, H.Y. and C.Z.; formal analysis, J.W., X.Y., and S.H.; investigation, W.X. and S.H.; writing—original draft preparation, J.W. and S.H.; writing—review and editing, H.Y. and C.Z.; visualization, J.W. and S.H.; supervision, H.Y. and C.Z. All authors have read and agreed to the published version of the manuscript.

**Funding:** This research was funded by Agriculture and Social Development Research Project of Hangzhou (202203A07), the Science Technology Department of Zhejiang Province (LGN20C140002), and Joint-extension Project of important Agriculture Technology in Zhejiang Province (2021XTTGSC02-4), the Postdoctoral Science Foundation of Zhejiang Province, China (ZJ2021121).

**Institutional Review Board Statement:** Not applicable.

**Informed Consent Statement:** Not applicable.

**Data Availability Statement:** Not applicable.

**Conflicts of Interest:** The authors declare no conflict of interest.

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
