# Peer review of "Involvement of CYP51A and CYP51B in Growth, Reproduction, Pathogenicity, and Sensitivity to Fungicides in Colletotrichum siamense"

_agronomy, doi:10.3390/agronomy13010239_

Round 1

Reviewer 1 Report

In this study, the authors assessed which type of pathological fungi is the dominant parasite causing Strawberry crown rot disease in greenhouses in Jiande, Zhejiang Province, China, and also assessed involvement of CYP51A and CYP51B genes in growth, reproduction and pathogenicity of Colletotrichum siamense strain and its sensitivity to azole class fungicides.

Despite the rather interesting data obtained by the authors, there are a number of comments and questions:

1) line 38-39: “Colletotrichum siamense is an important semi-biological nutritive fungal pathogen that affects a wide range of crops”

How can any pathogen be “semi-biological”? Could the authors explain what exactly they implied?

2) Line 44: “P450 sterol 14A-demethylase”

The common name for cytochromes of P450(51) subfamily is “sterol name 14-alpha-demethylases”. For example, “Lanosterol 14-alpha demethylase” of Candida albicans (UNIPROT P10613). I would recommend to use the “14-alpha-demethylases” or “14-α-demethylases” term.

3) Line 52: “Most ascomycetes carry two copies of CYP51A and CYP51B

It means that “Most ascomycetes carry two copies of CYP51 gene: CYP51A and CYP51B” or each of CYP51 genes has another copy on the fungi genome? I would recommend to clarify this sentence to prevent misunderstanding.

4) Line 79: "123°10’ E) from July to September 2021."

The coordinate line of “123°10’ E” corresponds to the sea area and does not match the border of the Jiande district of Zhejiang Province.

5) Figure 4 title “Figure 4. Colony morphology of wild-type JD-A-12 and CsCYP51A and CsCYP51B deletion mutants on PDA (a1–c1, a2–c2), conidia (a3–c3) and appressoria (a4–c4), scale bar: 236 (a3–c3, a4–b4) = 20 μm.” does not indicate which horizontal line corresponds to which strain of the fungus. I would recommend to revise the title.

6) Line 225-226: “The colony morphology of the CsCYP51A deletion mutants was not significantly 225 different from that of wild-type JD-A-12 on PDA plates.”

The mutant strains were obtained using the control JD-A-12 as a base strain? Why for the deletion experiment the JD-A-12 was chosen of all strains isolated?

7) Lines 301-302: "CsCYP51A and CsCYP51B deletion mutants to nine DMI fungicides, which allowed us to assess the binding affinity of DMIs to the CYP51 gene of C. siamense."

I disagree with this statement. Measurement of affinity involves recording the direct fact of interaction by such methods as surface plasmon resonance (doi: 10.3390/molecules26082237), spectral titration (doi: 10.1016/j.jmb.2010.01.075), isothermal titration calorimetry (doi: 10.1074/jbc.M708734200) etc. allowing to obtain the data of binding kinetic and affinity, complex half-life and binding energy (doi: 10.7554/eLife.57264) or at least by accessing the computed models of binding if the protein structural data is available (doi: 10.3389/fmolb.2020.586540 ).

8) It is not entirely clear why the authors isolated 74 strains of pathogenic fungi. The data obtained showed that the most common pathogen causing Strawberry crown rot at sampling sites was Colletotrichum siamense. This data is not new, because the authors refered to Zhang et al. (doi: 10.3390/jof8111161) where it was shown that “In China, C. siamense is the main pathogen of strawberry 37 crown rot”. This experiment is relevant to the issue of studying the effect of CYP51 genes on the action of fungicides only in that one of the isolated strains of the fungus was chosen as the basis for obtaining mutants with a gene deletion.

9) Could authors elaborate the discussion of the differs in pathogenicity of the knock-out mutants? It seems to me, that it would be interesting to discuss which of the CYP51 genes of Colletotrichum siamense are expressed at which stages of fungal growth or pathological process if this data is available. It would allow to draw conclusions about the mechanisms of action of antifungal drugs and their effect on mutant strains.

Are there any data on the difference in the expression products of CYP51A and CYP51B genes, if any, and the functions they perform?

10) I encourage the authors to revise the Abstract section according to the Nature template to make the study’s architecture more understandable (https://www.nature.com/documents/nature-summary-paragraph.pdf).

Reviewer 2 Report

Numerous comments have been made in the attached file that should be taken into account. The most important aspect to improve are the results presented on 3.5. (Effects of CYP51 Gene Deletion on the Sensitivity of C. siamense to DMIs). In some experiments it is not detailed how the statistical analysis was carried out.

The discussion is poor, it describes again the results and includes few bibliographical references. A greater effort should be made to discuss the central aspects of the work, with up-to-date references as a base.

Round 2

Reviewer 1 Report

I accept Authors' corrections and responses. I have no further critical comments.

Being the non-native english speaker i cannot judge about writing and style, but i have a feeling that language editing could be required.

Reviewer 2 Report

The manuscript has improved remarkably. In some sentences added in the introduction english should be carefully revised. Some observations are made in the attached file.

I still consider Figure 6A not representative of the results shown in Figure 6B for the mutant CsCYP51B. In figure 6B the mutant CsCYP51B is significantly less sensitive to all fungicides,  which is not visualized in figure 6a.

Some observations are made in the attached file.

I still consider Figure 6A not representative of the results shown in Figure 6B for the mutant CsCYP51B. In figure 6B the mutant CsCYP51B is significantly less sensitive to all fungicides,  which is not visualized in figure 6a.
